# Nutrition Literacy Among University Students in Beijing: Status, Determinants, and Implications

**DOI:** 10.3390/nu17233748

**Published:** 2025-11-28

**Authors:** Wenpeng Li, Bohao Yang, Jianrui Zhai, Jiahui Li, Lunrongyi Tian, Meihong Xu

**Affiliations:** 1The School of Public Health, Peking University, Beijing 100191, China; 2210306221@stu.pku.edu.cn (W.L.);; 2Department of Nutrition and Food Hygiene, School of Public Health, Beijing Key Laboratory of Toxicological Research and Risk, Assessment for Food Safety, Peking University, Beijing 100191, China; 3Institute of Medical Technology, Peking University Health Science Center, Beijing 100019, China

**Keywords:** nutrition literacy, university students, influencing factors, health behaviors

## Abstract

Background: Nutrition literacy (NL) plays a crucial role in shaping long-term health behaviors among college students, particularly during the transformative final phase of their school education. This study investigated the level of NL among college students in Beijing and examined its association with these behaviors. Methods: A cross-sectional online survey was conducted among 765 students from 12 universities in Beijing. The questionnaire comprised three sections: demographic characteristics, lifestyle factors, and a nutrition literacy scale (Cronbach’s α = 0.893; χ^2^/DF = 4.750; RMSEA = 0.048; GFI = 0.891; AGFI = 0.876). The NL scale was divided into two domains: cognition and skills. Descriptive statistics were used to summarize NL scores and their distributions across dimensions and subgroups. Group differences for categorical variables were examined using chi-square or Fisher’s exact tests. Logistic regression analyses was employed to identify factors associated with NL. Mediation effects were tested using the Baron and Kenny approach. Results: The mean NL score was 67.74 ± 9.07, with only 7.6% of participants achieving an excellent NL level. Several lifestyle factors were significantly associated with excellent NL. Students with monthly living expenses of 2000–3000 CNY (OR = 2.35, *p* = 0.019) and >3000 CNY (OR = 3.22, *p* = 0.023) had higher odds of excellent NL compared to those spending <2000 CNY. Occasional exercise (OR = 2.36, *p* = 0.026) and daily breakfast consumption (OR = 2.76, *p* = 0.027) were also positively associated with excellent NL. In contrast, frequent midnight snacking significantly reduced the likelihood of excellent NL (OR = 0.20, *p* = 0.031). Better self-rated health status was strongly correlated with higher NL (OR = 2.82, *p* = 0.012). Moreover, NL mediated the relationship between lifestyle factors and healthy eating behaviors, underscoring a gap between nutritional knowledge and practical food skills. Conclusions: Current findings indicated suboptimal nutrition literacy among college students in Beijing, particularly in food selection skills. Excellent NL rates were associated with demographic and lifestyle factors, with NL serving as a mediator between lifestyle and health-related behaviors. These results emphasize the need for targeted nutrition education programs to enhance both knowledge and practical skills among university students.

## 1. Introduction

According to the Global Burden of Diseases (GBDs) 2021 report, nutritional deficiencies globally resulted in approximately 48.9 million disability-adjusted life years (DALYs) and 222,000 deaths, emphasizing the critical role of nutrition in maintaining human health [1]. Additionally, increasing evidence suggested a considerable correlation between the rise in diet-related diseases and a general lack of food knowledge and skills among populations. Enhancing nutritional knowledge, maintaining optimal dietary behaviors, and managing a healthy weight were considered key modifiable factors for health promotion and chronic disease prevention [2,3].

Nutrition literacy (NL) was first defined by the National Library of Medicine in the United States in 2000 as “the ability of an individual to obtain, process, and understand basic health information and services to make appropriate health decisions” [4]. Currently, nutrition literacy was the degree to which a person can obtain, process, and grasp basic dietetic information and services to make healthy food choices [5]. In terms of conceptualization, recent research often drew on Nutbeam’s tripartite model of health literacy (functional, interactive, and critical NL) [6,7]. In the field of operationalization, commonly used tools were currently scales. Influenced by different cultural backgrounds and age-specific cognitive patterns, the emphasis of scales developed for different populations varies. For instance, ANLS [8] and INFOLIT [9] were designed for assessing adolescent nutritional literacy, while NHLS [10] and FNLQ [11] were tailored for adults. Studies have demonstrated a profound interaction between an individual’s NL and environmental factors. For example, the impact of family dietary habits and economic status during childhood [12], effective interventions during school years, etc. [13]. Current research based on the theory of planned behavior suggests that nutrition literacy could improve informed decision-making ability, thereby influencing health behaviors and reducing the risk of nutrition-related diseases such as obesity, type 2 diabetes, and cardiovascular diseases [14]. Improving NL levels could promote healthier eating habits, thereby enhancing dietary quality and ultimately affecting individual health outcomes, the quality of nutrition and health services, and healthcare investments [15].

College students represent a demographic at a crucial transitional phase, where the resulting dietary habits can influence long-term health outcomes, such as obesity and related noncommunicable diseases. This period offers a strategic opportunity for impactful nutritional interventions [16]. Despite the importance of this phase, studies showed wide disparities in NL among university students globally, ranging from high levels of functional NL among Norwegian students to considerably lower levels in China’s transient and rural populations [17,18,19,20]. These variations highlighted the need for assessing NL in different populations, as understanding the factors influencing NL can aid in the development of targeted interventions. By equipping college students with the necessary knowledge and skills to navigate food environments, we can empower them to make healthier decisions, ultimately reducing the risk of diet-related diseases and promoting long-term well-being [21,22,23].

Previous studies had predominantly focused on the level of NL and its influencing factors, with limited attention to behavioral aspects—an area that may exert interactive effects [16,17,21,24]. Moreover, these studies suffered from sampling biases that constrain the generalizability of their findings, compounded by challenges such as small sample sizes and inadequate representation of target populations. To bridge these gaps, our study conducted a comprehensive survey across multiple universities in Beijing, aiming to capture a more representative snapshot of college students’ NL. Notably, this research distinguished itself through its larger sample size and simultaneous exploration of NL’s mediating role. By examining the interplay between NL, demographic characteristics, and health behaviors, we aimed to generate evidence that informs the development of more effective strategies for enhancing nutritional education and fostering healthier lifestyles among this critical young population.

## 2. Materials and Methods

### 2.1. Study Design and Participants

This study was designed as a cross-sectional survey. The research commenced with the development of a preliminary questionnaire, which was refined through consultations with academic advisors and a presurvey involving 53 participants across a university in Beijing. Feedback from this presurvey was instrumental in further modifying the questionnaire. The finalized questionnaire was disseminated across 12 universities in Beijing to collect and analyze the data.

This study strictly adhered to medical research ethics guidelines. It had been approved by the Biomedical Ethics Committee of Peking University Health Science Center (Ethics Review Approval No.: IRB00001052-24012, approved on 20 March 2024). For university students, an embedded electronic informed consent module was integrated into the online questionnaire distributed via the Wenjuanxing platform (Ranxing Information Technology Co., Ltd., Changsha, China). Participants were required to check the ‘Read and agree to the informed consent form’ option before proceeding to the questionnaire completion. All participants were informed of their right to withdraw from the study at any time without affecting academic evaluation. Data was stored and analyzed anonymously with strictly restricted access permissions.

A convenience sampling method was employed. We used WeChat tools to distribute the questionnaire via the Wenjuanxing platform from April to May 2024 and reminded them to participate in this survey voluntarily. The inclusion criteria were as follows: (1) enrolled students at universities in Beijing and (2) undergraduate and junior college students. The exclusion criterion was as follows: participants could withdraw at any time, and their data were excluded.

### 2.2. Measures

#### 2.2.1. Sociodemographic, Lifestyle, and Health-Related Data

The questionnaire design for this study was structured into two main sections to assess NL among university students. The first section collected essential demographic and lifestyle information: sex, nationality, age, educational level, major, grade, and living expenses; and physical data such as body height and weight, which are necessary for calculating BMI. Lifestyle habits included alcohol consumption, smoking habits, general health conditions, exercise routines, takeaway food preferences, midnight snack consumption, and breakfast habits. Additionally, the survey explored various paths through which students acquired health knowledge, such as formal education, online sources, family, or peers. This broad range of data allowed for a nuanced analysis of how personal backgrounds, health behaviors, and educational experiences influence NL and dietary choices.

#### 2.2.2. Nutrition Literacy Data

The second part of the questionnaire focused on the detailed assessment of NL, using the China Adult Food and Nutrition Literacy Questionnaire (FNLQ) developed by the Department of Nutrition and Food Hygiene at the School of Public Health, Peking University. The questionnaire had been rigorously validated with good reliability and validity (Cronbach’s α = 0.893, χ^2^/DF = 4.750, RMSEA = 0.048, GFI = 0.891, AGFI = 0.876) [11,25]. FNLQ divided NL into two major categories—cognitive (knowledge) and skills (practice)—based on functional, interactive, and critical foundations. The cognitive section assessed participants’ knowledge and understanding of food nutrition concepts (such as food classification and healthy eating principles), while the skills section measured practical abilities, including food selection, intake (eating), and preparation (e.g., label reading, cooking methods). The questionnaire consisted of 50 questions with a total score of 100, where the knowledge dimension accounted for 8 points (Cronbach’s α = 0.866), and the practice dimension was further divided into food selection (30 points, Cronbach’s α = 0.845), food preparation (22 points, Cronbach’s α = 0.812), and eating (40 points, Cronbach’s α = 0.816). Previous studies had defined NL scores of 80 or higher as an “excellent” level. In addition, the questionnaire development process was completed through literature review, expert consensus, and two rounds of content validity testing to ensure its cultural adaptability for Chinese adults.

Postsurvey, adjustments to covariates were made on the basis of preliminary analysis, ensuring the accuracy and relevance of the findings. The complete version of the questionnaire was available in Supplement S1 of the publication for detailed review and replication of the study methodology.

Additionally, the following audit criteria were used for the collected questionnaires: (1) completion time under 150 s; (2) failure of quality inspection questions; and (3) exclusion of extreme body mass index (BMI) values as per the “Chinese Adult Overweight and Obesity Prevention and Control Guidelines,” specifically excluding those with a BMI > 61.5 or <13.9 [26].

### 2.3. Statistical Analysis

Considering the pre-survey 12% excellent NL rate and aiming for a 95% confidence interval with a width of 5%, the necessary sample size was calculated to be 688 individuals via PASS 2021 software. Adjusting for a 90% expected validity rate of the questionnaire, we targeted 765 university students for participation.

All the statistical tests were performed in R4.3.1. Descriptive statistics were present for NL scores, distributions across different dimensions, and various groups. Data were reported as categorical data with percentages and frequencies, quantitative variables as mean and standard deviation. To convert the scores of different NL dimensions into a total score out of 100, first convert each dimension’s raw score to a percentage of its respective raw full score, then weight and sum these percentages equally to obtain the standardized total score on a 100-point scale. A chi-square test or Fisher’s exact test was used to examine whether the distributions of categorical variables were significantly different between the groups. To compare score differences in different domains of nutritional literacy, we used the following statistical methods: for comparisons between cognitive and skills domains, we used the Wilcoxon sign rank test; for comparisons between three skill subdomains, we used the Friedman test. We used logistic regression analysis to identify factors associated with excellent NL (binary outcome: total NL score ≥ 80 vs. <80). The dependent variable was defined based on the validated threshold of the NL scale [11]. Subdomain scores were not used as dependent variables in regression models but were included in descriptive comparisons. Model results were reported as odds ratios (ORs) with 95% confidence intervals (CIs). For completeness, linear regression results using the continuous NL score as the dependent variable were provided in Appendix A. A two-tailed *p* value < 0.05 was considered statistically significant.

For the mediating effect, Baron and Kenny’s procedure was used to evaluate the mediating variables [27]. A variable was considered a mediating variable if it meets the following three conditions: (1) the independent variable affected the dependent variable, indicating a significant total effect; (2) the independent variable affected the mediating variable; and (3) both the independent variable and the mediating variable affected the dependent variable, and the effect of the mediating variable was significant. If the three regression models were robust, then a mediating effect exists. The significance of this indirect effect was subsequently verified via the bootstrap method [28]. By resampling, a bias-corrected (BC) confidence interval was constructed for the indirect effect. If the 95% confidence interval did not exceed 0, it indicated the existence of an indirect effect (or confirms the hypothesis of a mediating effect), and the ratio of the indirect effect to the total effect was calculated. The mediating analysis of this study aimed to explore the statistical path relationship between variables, and provided a hypothesis framework for the subsequent longitudinal study, without inferring causality.

## 3. Results

### 3.1. Demographic Characteristics of the Study Participants

A total of 823 participants completed the online questionnaire, 765 of whom met the quality inspection requirements; the valid questionnaire rate was 93.0%. In our study population, there were more females than males, accounting for 63.66% of the sample; medical students, 14.64%; freshmen, 33.20%; and key universities, 63.79%. The details are shown in Table 1.

### 3.2. Nutrition Literacy Status

The mean NL score for the participating college students was 66.74 ± 9.07. The rate of excellent comprehensive nutritional literacy was 7.6%. Among the NL-specific dimensions, the score for nutritional knowledge was 86.67 ± 16.88, and the excellent rating was 73.3%. For food selection, the scores were 65.15 ± 12.18 and 11.8%; for food preparation, the accuracies were 66.29 ± 12.51 and 12.6%; and for food intake, the accuracies were 66.70 ± 10.23 and 10.1%. Among the four dimensions, the nutrition knowledge cognition level of Chinese college students was relatively high, whereas the level of the three food skill dimensions was relatively low, especially for food selection skills. Notably, the gap between the scores of the excellent and non-excellent groups was the largest in the food selection dimension. Figure 1 shows the distribution of scores.

### 3.3. Relationship of NL with the Other Variables

For the association of NL with covariates, we first reported the univariate results via the chi-square test, with the specific results shown in Supplement Appendix A. Next, we reported the multivariate results via logistic regressions to identify factors associated with excellent NL (Table 2). For reference, linear regression results using the continuous NL score are provided in Appendix A.

In our results, in terms of the proportion of students demonstrating excellent NL, significant associations were found with several factors. Medical students were more likely to achieve excellent NL, with an OR of 2.44 (95%CI: 1.03–5.77, *p* = 0.042). Compared with students with monthly living expenses below 2000 CNY, those with incomes ranging from 2000 to 3000 CNY and above 3000 CNY were associated with increased NL, with ORs of 2.35 (95%CI: 1.03–5.77, *p* = 0.019) and 3.22 (*p* = 0.023), respectively. Furthermore, engaging in occasional exercise and consuming breakfast daily were positively correlated with higher NL rates, with ORs of 2.36 (*p* = 0.026) and 2.76 (*p* = 0.027), respectively. Self-reported stomach disease and overall health status also demonstrated significant positive associations with NL, with ORs of 1.96 (*p* = 0.039) and 2.82 (*p* = 0.012), respectively. Additionally, diverse personal methods of acquiring nutritional knowledge were strongly correlated with higher NL rates, with an OR of 2.45 (*p* = 0.015). In contrast, students from important schools, occasional food intake frequency and frequent snack frequency significantly associated with the lower NL rate likelihood, with ORs of 0.45 (*p* = 0.034), 0.19 (*p* = 0.025), and 0.20 (*p* = 0.031), respectively.

### 3.4. Mediating Effect of NLs on Health Behaviors

This study used the above data to further explore the mediating effect of NL on the relationship between multilevel factors and health-related behaviors among college students, as suggested by previous research [29]. Initially, we employed the chi-square test to establish the correlation between general factors and the healthy and eating behavior factors collected in this study and health-related behaviors. The results of this test are detailed in Supplement Appendix A. Building on these findings, we proceeded to analyze the role of NL as a mediator. We configured NL as the mediating variable, with general factors as independent variables and health-related behaviors as dependent variables. This analysis aimed to determine the extent to which NLs mediate the association of demographic and lifestyle factors on healthy behaviors. The mediation model followed the schematic framework illustrated in Figure 2. Specifically, for each pair of independent variables (e.g., major, living expenses) and dependent variables (e.g., breakfast frequency), NL was tested as a single mediator. Table 3 presents the detailed results of these mediation analyses, including effect sizes and association ratios. The analysis confirmed a significant mediating effect, indicating that NL does indeed mediate the relationship between general demographic factors and health behaviors. Taking living expenses as an example, our analysis reveals that NLs mediate 11.27% of the effect on exercise frequency and 16.28% of the effect on the frequency of takeout meals. These findings highlight the critical role of NLs as a mediator in shaping health-related behaviors.

## 4. Discussion

To the best of our knowledge, this study was the first to report the current status of NL among college students in Beijing, analyzing the associations among NL, general demographic characteristics, health status factors, and dietary behaviors. Additionally, it explored the potential mediating effect of NLs on these relationships, with a particular focus on the influence of university level and late-night eating frequency, which were factors that have not been extensively studied before.

According to our survey results, the overall excellent NL rate among Beijing college students was 7.6%. This exceeded the 6.2% rate reported for rural residents in Beijing and the 7.1% rate reported for primary and secondary school students in the region [20,30]. Our findings indicated that while college students generally possess good nutritional knowledge, their practical food skills, particularly in terms of food selection, were substantially weaker. This deficiency aligned with trends observed among rural residents in Beijing. These insights underscore the need for educational interventions that focus on enhancing practical food-related skills, especially food selection abilities, among college students.

College students are in the so-called “psychological weaning period,” with distinct nutritional literacy characteristics at this stage. First, this stage denotes a transition from passive reception of nutritional information to active knowledge-seeking. Students typically initiate independent searches for dietary guidelines and utilize campus health platforms or social media to acquire nutritional information, reflecting proactive exploration of healthy eating behaviors [31]. Second, the campus environment serves as a key context for autonomous food decision-making. Studies indicate that college students residing independently (in dormitories or outside parental households) tend to make more autonomous food choices, with their intake of fruits, vegetables, and meat significantly modulated by environmental factors, including campus cafeterias and nearby fast-food establishments [31,32]. Third, the university period is a critical window for establishing healthy behavioral patterns. Longitudinal tracking data show that although adolescents aged 14–21 exhibit quantitative changes in the consumption frequency of fruits, vegetables, and sugary drinks, their relative rankings remain stable during this phase—indicating that dietary patterns formed at this stage possess strong persistence [33]. In sharp contrast, adult dietary patterns are stabilized, with diminished plasticity in nutritional behaviors, often leading to less pronounced intervention effects than those in the university period.

Significant disparities in NL between Chinese individuals and those from other countries were also observed, particularly in terms of knowledge, skills, and behaviors. NL among Chinese students tends to progressively decline across these categories, highlighting a gap in effectively internalizing knowledge and converting it into practical skills and actions [19]. In contrast to many universities abroad, most educational institutions in China offer nutrition courses predominantly to students in specialized fields such as food science or medicine. This practice limits broader access to comprehensive nutrition education for students in nonspecialized disciplines. This restriction highlights the urgent need for curriculum reforms to extend nutrition education across all fields of study [19,34].

In terms of factors influencing NL, the literature suggests that medical students, seniors, and students with good self-reported health, higher living expenses, and a lower frequency of takeout consumption tend to have higher NL [16,23,35,36,37,38]. Our study, which is consistent with previous findings, revealed that students with medical majors, better self-reported health, multiple knowledge acquisition sources, higher living expenses, less frequent takeouts, and regular breakfast habits were more likely to have higher NL. Interestingly, this study was the first to report that students from higher-tier universities may have lower NL, contrary to our initial hypothesis. It is speculated that students from these institutions may face greater academic pressure, leading them to neglect nutritional knowledge. Although some studies have shown that students with better academic performance tend to have higher NL, these findings are based on single-institution studies and may not directly contradict our results [16].

This study is the first to identify an association between late-night snacking and NL, indicating that the frequent consumption of late-night snacks is correlated with lower NL. This relationship may stem from the connection between late-night snacking and unhealthy lifestyle habits, such as inadequate sleep. Additionally, while some research posits that an appropriate BMI correlates with higher NL, other studies argue that there is no significant relationship between BMI and NL. Our findings align with the latter perspective, suggesting that BMI may not be a reliable predictor of NL [24,38,39,40].

With respect to the mediating role of NL, previous research has explored its effect on health-related behaviors but often has not focused on specific behaviors [29,41]. In this study, we used specific behaviors—breakfast consumption, late-night snacking, takeout consumption, and exercise frequency—to verify the mediating effect of NL. The results suggest a possible mechanism that higher nutritional literacy may promote the application of nutritional knowledge, while enhancing multi-level factors related to healthy eating behaviors may simultaneously improve both nutritional literacy and health outcomes.

Despite its contributions, this study has several limitations. First, the convenience sampling method and relatively small sample size may introduce bias. Second, the use of self-reported data could lead to discrepancies between participants’ responses and their actual behaviors, potentially affecting the accuracy of the results. Third, this study is cross-sectional in nature, which limits our ability to establish causal relationships between variables in a strict sense. Therefore, our mediation analysis is based on hypotheses informed by the researchers’ knowledge, which may introduce uncertainties. Additionally, the cross-sectional design of this study cannot meet the temporal requirements of causal mediation analysis. Therefore, the results of mediation analysis should be interpreted as statistical associations between variables rather than causal mechanisms. These findings primarily provide hypotheses for future longitudinal studies. Nonetheless, the study employed a rigorous design, a validated scale, and included over 700 participants from 12 universities in Beijing. Additionally, we applied strict statistical procedures to minimize bias.

The association of factors such as BMI, sex, and grade on NL remains controversial. Future research should focus on these aspects, employing more robust random sampling methods and objective measurement tools to obtain more reliable results. Additionally, while most current studies are cross-sectional, future research could benefit from longitudinal or prospective designs to provide higher levels of evidence.

## 5. Conclusions

This study assessed NL among college students in Beijing, which is higher than the local average. The key factors associated with NL included living expenses, the consumption of snacks at night, exercise frequency, self-reported health status, and breakfast habits. Notably, practical food skills, particularly food selection, were found to be less developed than nutritional knowledge. Furthermore, NL may mediate the relationships between general factors and healthy and eating behavior factors. Our findings underscore the need for enhanced nutrition education and targeted interventions within universities, and larger and longitudinal studies are needed to enhance these findings.

## Figures and Tables

**Figure 1 nutrients-17-03748-f001:**
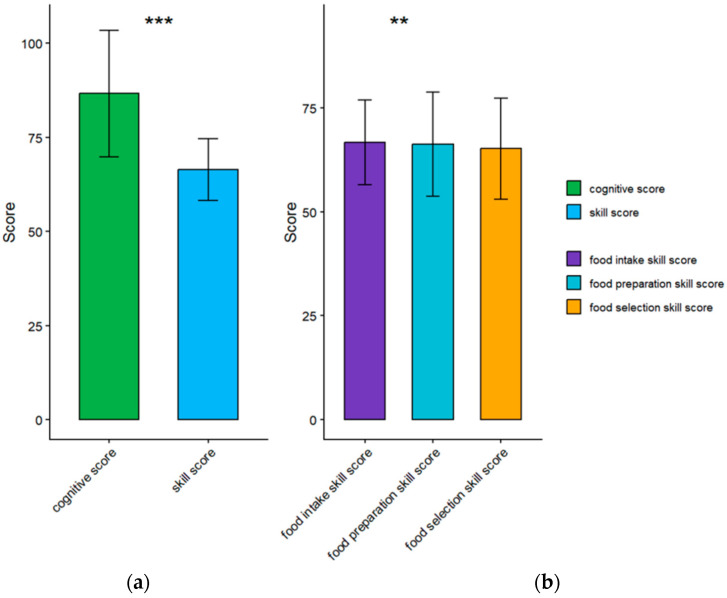
Score Distribution and Differences in all Dimensions of NL. (**a**) The difference between cognition and skill scores in the two dimensions of NL; (**b**) the difference between the three different skill dimensions. ** *p* < 0.01, *** *p* < 0.001.

**Figure 2 nutrients-17-03748-f002:**
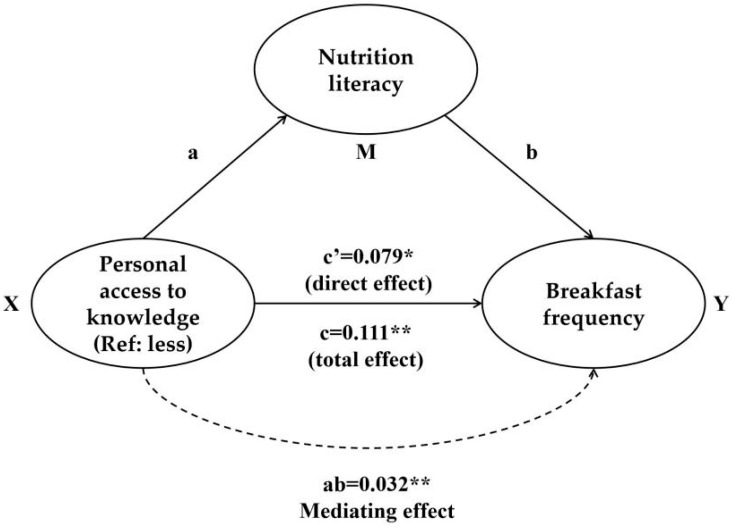
Schematic representation of the mediating effect. Note: In the mediation model, X is the independent variable, Y is the dependent variable, and M is the mediator variable, a denotes the effect of X on M, b denotes the effect of M on Y, c is the total effect of X on Y, c’ is the direct effect of X on Y after controlling for M, and ab is the indirect effect via M. * *p* < 0.05, ** *p* < 0.01.

**Table 1 nutrients-17-03748-t001:** Demographic characteristics of the participants.

Variable	Overall (N = 765)	Variable	Overall (N = 765)
Demographic Information		Personal ways of acquiring knowledge (types)	
Gender (Female)	487 (63.66)	High school	349 (45.62)
Major (Medical)	112 (14.64)	University lectures	260 (33.99)
Grade (Freshman)	254 (33.20)	College electives	263 (34.38)
Economic and Physical Data		Community and extracurricular activities	259 (33.86)
School Level (Elite)	488 (63.79)	There is almost no way	164 (21.44)
Living expenses (CNY/month)		Health-related behaviors	
0–2000	408(53.33)	Exercise frequency	
2000–3000	284(37.12)	Almost never	564 (73.73)
3000 and above	73 (9.54)	Occasionally	140 (9.54)
BMI		Often	86 (11.24)
Normal (18.5 ≤ BMI ≤ 24)	476 (62.22)	Takeaway frequency	
Overweight (BMI > 24)	171 (22.35)	Almost never	30 (3.92)
Thin (BMI < 18.5)	118 (15.42)	Occasionally	537 (70.20)
Lifestyle and Health status		Often	198 (25.88)
Drinking alcohol (Yes)	439 (57.39)	Frequency of midnight snacks	
Smoke (Yes)	26 (3.40)	Almost never	139 (18.17)
Self-reported health status (Good)	419 (54.77)	Occasionally	529 (69.15)
Gastrointestinal diseases (Yes)	281 (36.73)	Often	97 (12.68)
Gastrointestinal discomfort (Yes)	543 (70.98)	Breakfast frequency	
Knowledge Acquisition Paths		Almost never	332 (43.40)
Family suggestion		3–4 days per week	133 (17.39)
Almost never	52 (6.80)	5–6 days per week	185 (24.18)
Occasionally	375 (49.02)	Every day	215 (28.10)
Often	338 (44.18)		
Ways for families to acquire knowledge (types)			
Food and health magazines, books	327 (42.75)		
Short video (TikTok, Bilibili)	624 (81.57)		
Network nutrition class articles, push	478 (62.48)		
There is almost no way	42 (5.49)		

Note: Data are *n* (%) deviations. CNY: Chinese yuan. The classification of university levels is categorized into “Elite University” and “Standard University,” which are determined by whether the institution is included in China’s “Double First-Class” university initiative.

**Table 2 nutrients-17-03748-t002:** The association of variables on NL among college students.

Variable	Normal (N = 707)	Excellent (N = 58)	OR (95%CI)	*p*
Demographic Information				
Gender (Female: male as Ref.)	453 (93.02)	34 (6.98)	1.11(0.53–2.34)	0.787
Major (Medical: non-medical as Ref.)	97 (86.61)	15 (13.39)	2.44(1.03–5.77)	0.042
Grade (Freshman; non-Freshman as Ref.)	228 (89.76)	26 (10.24)	1.56 (0.80–3.05)	0.193
Economic and Physical Data				
School Level (Elite; Standard as Ref.)	458 (93.85)	30 (6.15)	0.45 (0.22–0.94)	0.034
Living expenses (CNY/month)				
0–2000	388 (95.10)	20 (4.90)	Ref.	Ref.
2000–3000	257 (90.49)	27 (9.51)	2.35 (1.15–4.80)	0.019
3000 and above	62 (84.93)	11 (15.07)	3.22 (1.18–8.80)	0.023
BMI				
Normal (18.5 ≤ BMI ≤ 24) as Ref	444 (93.28)	32 (6.72)	Ref.	Ref.
Overweight (BMI > 24)	152 (88.89)	19 (11.11)	1.99 (0.96–4.12)	0.065
Thin (BMI < 18.5)	111 (94.07)	7 (5.93)	1.06 (0.41–2.77)	0.903
Lifestyle and Health status				
Drinking alcohol: (Yes; never as Ref.)	412 (93.85)	28 (6.15)	0.77 (0.40–1.47)	0.426
Smoke: (Yes; never as Ref.)	25 (96.15)	1 (3.85)	0.41 (0.05–3.37)	0.404
Self-reported health status: (Good; not-good as Ref.)	371 (88.54)	48 (11.46)	2.82 (1.25–6.32)	0.012
Gastrointestinal diseases: (Yes; no as Ref.)	253 (90.04)	28 (9.96)	1.96 (1.03–3.74)	0.039
Gastrointestinal discomfort: (Yes; no as Ref.)	227 (93.42)	16 (6.58)	1.29 (0.62–2.68)	0.493
Knowledge Acquisition Paths				
Family suggestion				
Never as Ref	50 (96.15)	2 (3.85)	Ref.	Ref.
Occasionally	351 (93.60)	24 (6.40)	1.18 (0.23–6.04)	0.843
Often	306 (90.53)	32 (9.47)	1.34 (0.26–6.84)	0.726
Ways for families to acquire knowledge (types): (More; Less as Ref.)	172 (88.21)	23 (11.79)	1.16 (0.58–2.34)	0.656
Personal ways of acquiring knowledge (types): (More; Less as Ref.)	159 (85.48)	27 (14.52)	2.45 (1.19–5.03)	0.015
Health-related behaviors				
Exercise frequency				
Almost never as Ref	534 (94.68)	30 (5.32)	Ref.	Ref.
Occasionally	140 (88.05)	19 (11.95)	2.36 (1.11–5.02)	0.026
Often	33 (78.57)	9 (21.43)	2.31 (0.76–6.96)	0.138
Takeaway frequency				
Almost never as Ref	24 (80.00)	6 (20.00)	Ref.	Ref.
Occasionally	493 (91.81)	44 (8.19)	0.35 (0.11–1.19)	0.093
Often	190 (95.96)	8 (4.04)	0.19 (0.05–0.81)	0.025
Frequency of midnight snacks				
Almost never as Ref	118 (84.89)	21 (15.11)	Ref.	Ref.
Occasionally	495 (93.57)	34 (6.43)	0.48 (0.23–1.02)	0.057
Often	94 (96.91)	3 (3.09)	0.20 (0.05–0.86)	0.031
Breakfast frequency				
Almost never as Ref	224 (96.55)	8 (3.45)	Ref.	Ref.
3–4 days per week	125 (93.98)	8 (6.02)	1.73 (0.57–5.22)	0.330
5–6 days per week	177 (95.68)	8 (4.32)	0.79 (0.26–2.37)	0.675
Every day	181 (84.19)	34 (15.81)	2.76 (1.12–6.80)	0.027

Note: Data are *n* (%) or the mean ± standard deviation. CNY: Chinese yuan. The classification of university levels is categorized into “Elite University” and “Standard University,” which are determined by whether the institution is included in China’s “Double First-Class” university initiative. “Ways for families to acquire knowledge” and “personal ways of acquiring knowledge” were defined as less if there were fewer than three ways; otherwise, they were defined as more.

**Table 3 nutrients-17-03748-t003:** Test for the mediating effect of NL between general factors and health-related behavior factors.

Independent Variable	Dependent Variable	Effect Type	Effect Size	95% CI	*p* Value	Association Ratio (%)
Major (Ref: Non-medical)	Breakfast frequency	Mediating effect (ab)	0.014	[0.000, 0.026]	0.038	11.97
		Direct effect (c’)	0.103	[0.017, 0.193]	0.018	
		Total effect (c)	0.117	[0.031, 0.204]	0.007	
Grade (Ref: non-Freshman)	Exercise frequency	Mediating effect (ab)	0.016	[−0002, 0.027]	0.031	18.82
		Direct effect (c’)	−0.085	[−0.153, −0.021]	0.010	
		Total effect (c)	−0.069	[−0.145, −0.013]	0.042	
Living expenses (Ref: 0–2000 CNY/month)	Exercise frequency	Mediating effect (ab)	0.016	[0.001, 0.025]	0.008	11.27
		Direct effect (c’)	0.126	[0.065, 0.187]	<0.001	
		Total effect (c)	0.142	[0.075, 0.198]	<0.001	
Living expenses (Ref: 0–2000 CNY/month)	Takeaway frequency	Mediating effect (ab)	−0.007	[−0.014, −0.000]	0.040	16.28
		Direct effect (c’)	0.043	[0.014, 0.071]	0.003	
		Total effect (c)	0.035	[0.010, 0.065]	0.011	
Personal access to knowledge (Ref: less)	Breakfast frequency	Mediating effect (ab)	0.032	[0.015, 0.054]	0.001	28.83
		Direct effect (c’)	0.079	[0.007, 0.154]	0.035	
		Total effect (c)	0.111	[0.040, 0.186]	0.003	

Note: General factors were defined as demographic information, economic and physical data, lifestyle and health status and knowledge acquisition paths, and eating behavior factors were defined as dietary behaviors. In this table, we show all the results with significant mediation effects.

## Data Availability

The data presented in this study are available upon request from the corresponding author. The data are not publicly available due to privacy and ethical concerns; neither the data nor the source of the data can be made available.

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
