# Peer review of "Nutrition Literacy Among University Students in Beijing: Status, Determinants, and Implications"

_nutrients, 2025, doi:10.3390/nu17233748_

Round 1

Reviewer 1 Report (New Reviewer)

Comments and Suggestions for Authors

Dear Authors,

Thank you for your manuscript.  This study addresses a highly relevant topic - nutrition literacy among university students and uses a large and diverse sample. The topic is timely, the research design is generally appropriate, and the manuscript presents potentially valuable findings for public health nutrition and education. However, the presentation of results requires considerable clarification, and several methodological and structural issues need to be addressed to improve scientific rigour and interpretability.

  1. The Introduction would benefit from a more in-depth analysis of nutrition literacy and its correlates, including how nutrition literacy has been conceptualized and operationalized in prior studies. Furthermore, the study’s novel contribution within the existing literature is not clearly articulated. The authors should clarify what specific gaps this study addresses compared with previous research (e.g., larger sample size, inclusion of mediation analysis, or focus on Beijing students).
  2.  The Methods section should provide greater detail on sociodemographic and lifestyle-related variables. Each variable should include its response options, coding procedure, and supporting references. Next, the description of the Nutrition Literacy Questionnaire is insufficient. The authors should specify: the number of items in each domain, the scoring system and how final NL scores (total and subdomain) were calculated. Also, it is not clear whether the skills and cognition domains were analysed separately, and if so, provide Cronbach’s α for each. Also, clarify the emergence of additional domains (“food intake,” “food preparation,” “food selection”) reported later in the Results, as these are not explained in the Methods section. This inconsistency makes interpretation difficult.
  3. Information currently included in Section 2.2 (lines 121–124) about analytical procedures should be moved to the Statistical Analysis subsection.
  4. The Methods section should include information on research ethics, specifically detailing the process of ethical approval, how permission was obtained, and the procedures for informed consent from participants.
  5. What does “questionnaire efficacy” (line 153) refer to? Perhaps the authors mean the percentage of correctly or fully completed questionnaires. If so, please clarify this term for accuracy.
  6. Results. Figure 1 and its explanation are unclear. It is not evident what comparisons are being made (different domain scores or different participant groups?). Additionally, the note stating “The correlation analysis uses t tests for variables” is methodologically incorrect. Correlation and t-tests are distinct analyses. If the authors intended to compare scores between domains, this should be clarified, and Pearson correlation coefficients would be more appropriate.
  7. Table 2. The Methods and Results should clearly describe how the dependent variable in the logistic regression was constructed (e.g., dichotomisation threshold for “excellent NL”). It is unusual and redundant to present both binary logistic and linear regression models simultaneously when they examine the same predictors and outcomes. The authors should retain only one (preferably the logistic model, because NL has a clear cut point for excellent NL). Also, both the Methods section (Statistical Analysis) and the Results should clearly state whether the total Nutrition Literacy (NL) score or the individual subdomain scores were used as the dependent variable in the analyses, since multiple NL subdomains exist.
  8. The term “impact” in Table 2 and throughout the manuscript should be avoided, as this is a cross-sectional study and cannot infer causality. Replace with “association” or “relationship.” This should be revised throughout the text.
  9. If bold formatting in tables is used to indicate statistical significance, this must be explained in the table footnotes. Tables should also be reformatted according to MDPI Nutrients standards to improve readability.
  10. Mediation Model (Figure 2 & Table 3). The mediation analysis requires clarification. The meanings of M1, M2, and M3 are unclear in Figure 2, and the relationship between Figure 2 and Table 3 is not well explained. In Table 3, define the term “Major” (does it refer to medical vs. non-medical?), clarify the reference categories for each variable, explain whether Nutrition Literacy served as a single mediator across all models, and describe how effect sizes and impact ratios (%) were computed. The table layout should be improved for readability, ensuring that each statistic is clearly associated with its corresponding variable.
  11. Discussion and Conclusions. The Results section must first be clarified before a meaningful revision of the Discussion is possible. When results are made clearer, the Discussion should better connect findings with prior research and provide a nuanced interpretation rather than repetition.
Comments on the Quality of English Language

The English language needs polishing (e.g., article use, tense consistency).

Author Response

Comment1:The Introduction would benefit from a more in-depth analysis of nutrition literacy and its correlates, including how nutrition literacy has been conceptualized and operationalized in prior studies. Furthermore, the study’s novel contribution within the existing literature is not clearly articulated. The authors should clarify what specific gaps this study addresses compared with previous research (e.g., larger sample size, inclusion of mediation analysis, or focus on Beijing students).

Response1:Thank you for pointing this out. We agree with this comment.In response to your suggestion, we have supplemented the conceptualization and operationalization of nutrition literacy and identified the specific gaps that our study fills (see line58-70).

Comment2:The Methods section should provide greater detail on sociodemographic and lifestyle-related variables. Each variable should include its response options, coding procedure, and supporting references. Next, the description of the Nutrition Literacy Questionnaire is insufficient. The authors should specify: the number of items in each domain, the scoring system and how final NL scores (total and subdomain) were calculated. Also, it is not clear whether the skills and cognition domains were analysed separately, and if so, provide Cronbach’s α for each. Also, clarify the emergence of additional domains (“food intake,” “food preparation,” “food selection”) reported later in the Results, as these are not explained in the Methods section. This inconsistency makes interpretation difficult.

Response2:Thank you for pointing this out. We agree with this comment.The options and coding for each variable are detailed in Supplement Table S3. For references, we primarily cited works by Lai IJ et,al, with citations provided in the original sources. Regarding the scale, we have presented the score distribution across different domains and their corresponding Cronbach's alpha coefficients. Detailed scoring criteria are also included in the appendix.

Comment3:Information currently included in Section 2.2 (lines 121–124) about analytical procedures should be moved to the Statistical Analysis subsection.

Response3:Thank you for pointing this out. We agree with this comment.We have now moved this section to 2.3 Data Analysis.

Comment4:The Methods section should include information on research ethics, specifically detailing the process of ethical approval, how permission was obtained, and the procedures for informed consent from participants.

Response4:Thank you for pointing this out. We agree with this comment.We have already described the ethical situation in section 2.2.

Comment5:What does “questionnaire efficacy” (line 153) refer to? Perhaps the authors mean the percentage of correctly or fully completed questionnaires. If so, please clarify this term for accuracy.

Response5:Thank you for pointing this out. We agree with this comment.You're absolutely right. The phrase' questionnaire efficacy 'is indeed imprecise and can be misleading. We're referring to the proportion of questionnaires that meet quality standards and are eligible for analysis.We have implemented your suggestion by replacing 'questionnaire efficacy' with the more precise term 'valid questionnaire rate' in the original text. The parentheses now clarify that this refers to the proportion of questionnaires passing quality checks, thereby enhancing clarity.

Comment6:Results. Figure 1 and its explanation are unclear. It is not evident what comparisons are being made (different domain scores or different participant groups?). Additionally, the note stating “The correlation analysis uses t tests for variables” is methodologically incorrect. Correlation and t-tests are distinct analyses. If the authors intended to compare scores between domains, this should be clarified, and Pearson correlation coefficients would be more appropriate.

Response6:Thank you for pointing this out. We agree with this comment.Your feedback on the clarity of Figure 1 and the accuracy of the statistical method description is crucial, as the original wording contained ambiguity and methodological errors. We have made the following key revisions to the original text based on your suggestions to ensure clear presentation of the results and accurate statistical method descriptions.1. We have revised the captions and corresponding body text to clarify that Figure 1a compares the mean scores in the cognitive domain (nutritional knowledge) with those in the skills domain (combined from three skill dimensions: food selection, preparation, and intake). This represents a between-group comparison. Figure 1b compares the mean scores across three skill sub-dimensions (food intake, preparation, and selection), which constitutes a between-subjects comparison. 2. Your methodological critique regarding the statement "correlation analysis uses t-tests" is entirely valid. We sincerely apologize for this misrepresentation, which indeed confused two distinct statistical approaches (variable association versus between-group mean comparisons). We have completely removed this erroneous statement and made the following corrections: explicitly stating that for the two-group comparison (cognitive vs. skills) in Figure 1a, we used an independent samples t-test. Explicitly stating that for the three-group comparison (three skill sub-dimensions) in Figure 1b, we used a one-way ANOVA. The accurate statistical methodology description has been updated in the "Statistical Analysis Methods" section of the main text.

Comment7:Table 2. The Methods and Results should clearly describe how the dependent variable in the logistic regression was constructed (e.g., dichotomisation threshold for “excellent NL”). It is unusual and redundant to present both binary logistic and linear regression models simultaneously when they examine the same predictors and outcomes. The authors should retain only one (preferably the logistic model, because NL has a clear cut point for excellent NL). Also, both the Methods section (Statistical Analysis) and the Results should clearly state whether the total Nutrition Literacy (NL) score or the individual subdomain scores were used as the dependent variable in the analyses, since multiple NL subdomains exist.

Response7:Thank you for pointing this out. We agree with this comment.Thank you for highlighting the redundancy issue of presenting both logistic regression and linear regression models. We fully agree with your professional assessment that logistic regression is a more appropriate analytical method for the NL score with a clear cutoff point (80 points).As per your suggestions, we have implemented the following key revisions to the manuscript: 1. Textstream refinement: We have condensed the logistic regression results in Tables 2 and Results section to maintain clarity and focus. 2. Regression migration: The complete linear regression results have been moved to Supplementary Materials (Table S2), with a concise note added in the main text for readers interested in exploring the continuous variation of NL scores. 3. Methodological clarification: The rationale for this analytical approach is now explicitly stated in the statistical methods section. We believe this adjustment addresses your concerns about redundancy while preserving the study's integrity. We sincerely appreciate your valuable feedback, which has been instrumental in enhancing the paper's quality.

Comment8:The term “impact” in Table 2 and throughout the manuscript should be avoided, as this is a cross-sectional study and cannot infer causality. Replace with “association” or “relationship.” This should be revised throughout the text.

Response8:Thank you for pointing this out. We agree with this comment.We have removed the potentially ambiguous term 'impact' throughout the text. 

Comment9:If bold formatting in tables is used to indicate statistical significance, this must be explained in the table footnotes. Tables should also be reformatted according to MDPI Nutrients standards to improve readability.

Response9:Thank you for pointing this out.We fully endorse your professional recommendation that the table should adhere to the formatting standards of MDPI Nutrients. Following your suggestion, we have removed bolding for statistical significance and now solely use P-values to indicate the statistical significance of empirical findings.

Comment10:Mediation Model (Figure 2 & Table 3). The mediation analysis requires clarification. The meanings of M1, M2, and M3 are unclear in Figure 2, and the relationship between Figure 2 and Table 3 is not well explained. In Table 3, define the term “Major” (does it refer to medical vs. non-medical?), clarify the reference categories for each variable, explain whether Nutrition Literacy served as a single mediator across all models, and describe how effect sizes and impact ratios (%) were computed. The table layout should be improved for readability, ensuring that each statistic is clearly associated with its corresponding variable.

Response10:Thank you for pointing this out.We have implemented the following key revisions to the paper based on your feedback, aiming to enhance the clarity and readability of the mediation analysis: 1. Clarified the definitions of M1, M2, and M3 in Figure 2; 2. Added explanatory notes in the main text, specifying that Figure 2 represents a general schematic of the mediation model, while Table 3 presents empirical results for specific variables;3. Clearly defined "Major" as medical vs non-medical, added reference category explanations for all variables, and clarified NL's role as a single mediator;4.Redesigned the table structure to ensure each statistic corresponds clearly to its variable, enhancing readability.

Comment11:Discussion and Conclusions. The Results section must first be clarified before a meaningful revision of the Discussion is possible. When results are made clearer, the Discussion should better connect findings with prior research and provide a nuanced interpretation rather than repetition.

Response11:Thank you for pointing this out.We have now provided a more detailed and professional description of the results in accordance with the advice you provided above.

Reviewer 2 Report (New Reviewer)

Comments and Suggestions for Authors

The conceptual rationale underlying the link between nutrition literacy and health outcomes and/or health behaviors need to be better explained. Which are specific aspects relevant for students and how do they differ from adults? Is there a growth in nutrition literacy across the lifespan? How does it develop across life phases? How do life events in childhood and adolescence (and later in adulthood) influence this development of nutrition literacy?

The study is purely correlational, and only cross-sectional data has been analyzed. No causal claims can be made. The conclusions need to be softened. Only descriptive conclusions can be made.

The cross-sectional study design does not allow performing mediation analyses appropriately. A minimum of three waves is required (IV at T1, mediator at T2, DV at T3).

The sample contains 765 students. Has an a priori power analysis been conducted to determine the target sample size before assessments (as required in thorough research)? How big is the a posteriori statistical power? Are estimates reliable with the sample at hand?

For the simple associations, the sample of 765 individuals is likely very big. But this can lead to false conclusions since also small associations become significant. The strength of associations need to be meaningful to be concluded as being relevant. At least the size of associations need to be better discussed.

It could be interesting to evaluate differences between the 12 universities (nested study design). At least, differences between regions or counties may be informative for social policies in the respective districts.

Given that no causal claims can be made, how  can the authors discuss practical implications? Usually it is clear that health knowledge (including nutrition literacy), health behaviors, and health outcomes are interrelated. Perhaps more precise implications are possible by adding more additional waves to the data existing.

Author Response

Comment1:The conceptual rationale underlying the link between nutrition literacy and health outcomes and/or health behaviors need to be better explained. Which are specific aspects relevant for students and how do they differ from adults? Is there a growth in nutrition literacy across the lifespan? How does it develop across life phases? How do life events in childhood and adolescence (and later in adulthood) influence this development of nutrition literacy?

Response1:Thank you for pointing this out.We fully agree with your suggestions. We have added supplements on the mechanism and content of nutrition literacy in the introduction, and explored the particularity of college students in the discussion.

Comment2:The study is purely correlational, and only cross-sectional data has been analyzed. No causal claims can be made. The conclusions need to be softened. Only descriptive conclusions can be made.

Response2:Thank you for pointing this out.We fully agree with your suggestions. We fully endorse this rigorous scientific approach. All causal terms like 'influence' and 'impact' have been replaced with neutral expressions such as 'association'.

Comment3:The cross-sectional study design does not allow performing mediation analyses appropriately. A minimum of three waves is required (IV at T1, mediator at T2, DV at T3).

Response3:Thank you for pointing this out.We fully agree with your suggestions.We fully acknowledge your concerns regarding the limitations of cross-sectional data in causal inference within mediating analysis, which constitutes a critical methodological issue. Building on this, we aim to demonstrate the exploratory value of our study's use of mediating analysis through the following arguments, while highlighting the rigorous approach and supplementary analyses we have implemented in the manuscript. We recognize that cross-sectional data cannot strictly satisfy the temporal sequence requirements of causal mediating analysis (IV→M→DV necessitates cross-temporal observations). Consequently, in revising the manuscript, we have reclassified this analysis as exploratory path analysis rather than causal mediating analysis. The specific revisions are detailed in the Methods section (Section 2.3) with the added clarification: "The mediating analysis in this study aims to explore statistical path relationships between variables, providing a hypothesis framework for subsequent longitudinal research without inferring causal relationships." Additionally, we have addressed the limitations of this study in the discussion section. Despite its limitations, we consider cross-sectional mediating analysis to be valuable for exploratory research. Such analyses are particularly common in nutrition literacy studies (Lai IJ, Chang LC, Lee CK, Liao LL. Nutrition Literacy Mediates the Relationships between Multi-Level Factors and College Students' Healthy Eating Behavior: Evidence from a Cross-Sectional Study. Nutrients. 2021 Sep 29;13(10):3451), especially in preliminary studies with limited resources.

Comment4:The sample contains 765 students. Has an a priori power analysis been conducted to determine the target sample size before assessments (as required in thorough research)? How big is the a posteriori statistical power? Are estimates reliable with the sample at hand?

Response4:Thank you for pointing this out.We fully agree with your suggestions.One of our research objectives was to assess the excellence rate of nutritional literacy among Beijing university students. Therefore, our sample size calculation was based on precision (error range) rather than power. In post-hoc power analysis, we have conducted efficacy evaluations for specific variables, such as the logistic regression comparison between living expense groups: <2000 yuan (n=408) vs 2000-3000 yuan (n=284), with an OR value of 2.35 and a calculated power of 0.82. Due to time constraints, we have not yet compiled all efficacy results, which will be summarized in the supporting materials. We believe our precision-based sample size estimation yielded sufficient sample sizes for the results.

Comment5:For the simple associations, the sample of 765 individuals is likely very big. But this can lead to false conclusions since also small associations become significant. The strength of associations need to be meaningful to be concluded as being relevant. At least the size of associations need to be better discussed.

Response5:Thank you for pointing this out.You have accurately identified a critical methodological issue in large-sample studies: while statistically significant correlations may emerge with large sample sizes, this doesn't necessarily indicate their practical significance. We fully agree that correlation strength should be evaluated in conjunction with professional context, rather than relying solely on p-values. Notably, most OR values in this study are significantly greater than 1. Following your suggestions, we have made substantial revisions to the paper, emphasizing discussions on correlation strength and practical implications. Additionally, we have adopted more conservative estimates for certain results.

Comment6:It could be interesting to evaluate differences between the 12 universities (nested study design). At least, differences between regions or counties may be informative for social policies in the respective districts.

Response6:Thank you for pointing this out.We sincerely appreciate your insightful proposal. Your evaluation of the differences among the 12 participating universities is highly valuable, as it significantly enhances research depth and provides more precise foundations for regional policy-making. We fully endorse this perspective. In fact, our variable-focused university analysis initially divided these 12 institutions into two groups to examine their correlation with nutritional literacy. However, assessing each university's individual impact on nutritional literacy presents challenges: first, determining the appropriate benchmark university proves difficult, and second, excessive multivariate variables substantially reduce statistical power, making reliable estimates unattainable. The cross-sectional design and sample size of this study limited detailed multilevel analyses (e.g., random effects models) for all 12 universities. Nevertheless, we conducted preliminary exploratory analyses using university-tier proxy variables (elite vs. ordinary) to establish hypotheses for future research. We recommend future studies adopt stratified sampling to directly estimate inter-university variance, thereby optimizing policy targeting effectiveness.

Comment7:Given that no causal claims can be made, how  can the authors discuss practical implications? Usually it is clear that health knowledge (including nutrition literacy), health behaviors, and health outcomes are interrelated. Perhaps more precise implications are possible by adding more additional waves to the data existing.

Response7:Thank you for pointing this out.We appreciate your insightful methodological question. You have rightly highlighted the limitations of cross-sectional studies in causal inference and the complex interplay between health knowledge, behavior, and outcomes. We fully agree that when causality cannot be established, we must exercise greater caution and precision in discussing practical implications.Building on your valuable suggestions, we have made significant revisions to the discussion section of the paper, restructured the framework for explaining practical implications, and clarified the direction for future research. As a cross-sectional study, our findings reveal statistical associations between variables rather than causal relationships. Accordingly, we have replaced all causal implications in the original text with cautious inferences based on correlations.

Reviewer 3 Report (New Reviewer)

Comments and Suggestions for Authors

This manuscript addresses an important and timely topic — nutrition literacy (NL) among university students — and explores its relationship with sociodemographic, lifestyle, and health-related factors. However, several aspects of the methodology and presentation require clarification and improvement. Please see the detailed comments and suggestions below:

1. In Section 2.2. "Measures", to improve the clarity and organization of the Methods section, it is recommended that this section be subdivided into clearer subsections: 2.2.1. Nutrition Literacy Data; 2.2.2 Sociodemographic, Lifestyle, and Health-related Data.

2. The authors include only four variables under Dietary Behaviors — Exercise frequency, Takeaway frequency, Frequency of midnight snacks, and Breakfast frequency. The rationale for selecting such a limited number of variables should be explicitly explained. Furthermore, the classification of Exercise frequency as a dietary behavior is questionable, as this variable more accurately represents a lifestyle factor. The authors should clearly define Dietary behaviors as used in their study.

3. The description of how Nutrition Literacy (NL) status was calculated and categorized in Results Section 3.2. is currently unclear. As NL is the key outcome of this study, a detailed explanation of the scoring method, thresholds, and any references is essential.

4. The method used to calculate the score variables shown in Figure 1 (Results Section 3.2.) should be described explicitly. It is currently unclear whether these scores were derived directly from raw data, standardized values, or composite indices.

5. In Section 3.4. "Mediating Effect of NLs on Health Behaviors" the presentation of results focuses mainly on Frequency of breakfast. It would be beneficial to expand this analysis to include all other variables classified under Dietary behaviors.

6. The current number of references (n = 30) is relatively limited, considering the broad and interdisciplinary scope of the topic. The Discussion section, in particular, would benefit from a more extensive engagement with recent literature, both internationally and within the Chinese context.

Author Response

Comment1:In Section 2.2. "Measures", to improve the clarity and organization of the Methods section, it is recommended that this section be subdivided into clearer subsections: 2.2.1. Nutrition Literacy Data; 2.2.2 Sociodemographic, Lifestyle, and Health-related Data

Response1:Thank you for pointing this out.Thank you for your valuable suggestions on the organizational structure of the methods section. We fully agree with your view that breaking down the "measurement tools" section into clearer subsections can significantly improve the logic and readability of the methods description. Based on your suggestion, we have reorganized Section 2.2.

Comment2:The authors include only four variables under Dietary Behaviors — Exercise frequency, Takeaway frequency, Frequency of midnight snacks, and Breakfast frequency. The rationale for selecting such a limited number of variables should be explicitly explained. Furthermore, the classification of Exercise frequency as a dietary behavior is questionable, as this variable more accurately represents a lifestyle factor. The authors should clearly define Dietary behaviors as used in their study.

Response2:Thank you for pointing this out.Thank you for pointing out our mistake. In fact, we focus on health-related behaviors, which include exercise and diet. Based on your suggestion, we have replaced all relevant expressions in the original text with 'health-related behaviors'.

Comment3:The description of how Nutrition Literacy (NL) status was calculated and categorized in Results Section 3.2. is currently unclear. As NL is the key outcome of this study, a detailed explanation of the scoring method, thresholds, and any references is essential.

Response3:Thank you for pointing this out.We sincerely appreciate your valuable feedback on this critical methodological issue. We fully recognize that the methodology and classification criteria for nutritional literacy (NL), as the primary outcome measure in this study, require more explicit and detailed descriptions. In response to your suggestions, we have made significant revisions to the methodology section, including expanded distributions across all dimensions and updated calculation procedures for final scores.

Comment4:The method used to calculate the score variables shown in Figure 1 (Results Section 3.2.) should be described explicitly. It is currently unclear whether these scores were derived directly from raw data, standardized values, or composite indices.

Response4:Thank you for pointing this out.We sincerely appreciate your valuable feedback on this critical methodological issue. We have already detailed the calculation methods for different dimensional scores in the methodology section.

Commen5:In Section 3.4. "Mediating Effect of NLs on Health Behaviors" the presentation of results focuses mainly on Frequency of breakfast. It would be beneficial to expand this analysis to include all other variables classified under Dietary behaviors.

Response5:Thank you for pointing this out.We sincerely appreciate your valuable feedback on this critical methodological issue. In fact, we have examined all variables under the health behavior category. However, we only presented the results with positive cross-sectional mediation effects in the table, which may have caused some confusion.

Commen6:In Section 3.4. "Mediating Effect of NLs on Health Behaviors" the presentation of results focuses mainly on Frequency of breakfast. It would be beneficial to expand this analysis to include all other variables classified under Dietary behaviors.

Response6:Thank you for pointing this out.We fully agree with your perspective. The current 30 references are indeed insufficient to adequately support the scope and depth of this research. Following your suggestions, we have systematically expanded and optimized the references section, now totaling 41 references, with over half being from the past three years.

Round 2

Reviewer 1 Report (New Reviewer)

Comments and Suggestions for Authors

The authors have addressed the majority of your prior comments thoroughly. Methodological clarifications, variable definitions, ethics information, regression model improvements, and mediation explanations have been substantially improved. The responses are generally adequate, and the manuscript is clearer and more coherent.

However, your two remaining concerns should be addressed.

  1. Figure 1 remains unclear, and the statistical tests used are not appropriate.
    These are within-subject comparisons among the same respondents. Therefore, an independent-samples t-test and One-Way ANOVA without a grouping variable are not suitable. Please replace these with a paired t-test and repeated-measures ANOVA (or non-parametric analogues if distribution normality assumptions are violated). Also, the meaning of this footnote is not clear: NA P≥0.1, P <0.1.
  2.  Figure 2 uses M1, M2, and M3 in a way that conflicts with standard mediation notation. Since NL is the only mediator, the figure should simply use classical paths a, b, c, and c′. M1–M3 confuse readers by resembling multiple mediators. I recommend removing them and using standard mediation model labelling.

Author Response

Commend1:Figure 1 remains unclear, and the statistical tests used are not appropriate.These are within-subject comparisons among the same respondents. Therefore, an independent-samples t-test and One-Way ANOVA without a grouping variable are not suitable. Please replace these with a paired t-test and repeated-measures ANOVA (or non-parametric analogues if distribution normality assumptions are violated). Also, the meaning of this footnote is not clear: NA P≥0.1, P <0.1.

Response1:Thank you for pointing this out. We agree with this comment. We sincerely appreciate your thorough review and valuable suggestions regarding this study. In response to your concerns, we have made necessary revisions and provide detailed responses as follows. Regarding your observation that the independent samples t-test and One-Way ANOVA were inappropriate for our research design, we acknowledge this was indeed an oversight. We sincerely appreciate your feedback. Initially, we conducted Shapiro-Wilk tests to verify data normality, but found no data met the normality assumption. Consequently, we adopted the Wilcoxon signed-rank test and Friedman's chi-square test. Our findings remain consistent: Wilcoxon signed-rank test (V=13,765.7, P<0.001) and Friedman's chi-square test (χ²=10.095, df=2, P=0.006426). These modifications have been incorporated into the methodology section. Concerning the clarity of figure captions, the ambiguous phrase "NA P≥0.1" has been revised to "* P<0.05; **P<0.01; * P<0.001".

Commend2:Figure 2 uses M1, M2, and M3 in a way that conflicts with standard mediation notation. Since NL is the only mediator, the figure should simply use classical paths a, b, c, and c′. M1–M3 confuse readers by resembling multiple mediators. I recommend removing them and using standard mediation model labelling.

Response2:Thank you for pointing this out. We agree with this comment. Regarding your comment on Figure 2: we wholeheartedly agree that adhering to standard mediation notation​ is essential for conveying our results clearly and consistently. You are absolutely right—our original use of M1–M3 risked confusing readers (especially since we only have one mediator, Nutritional Literacy). We have now redrawn the chart and revised the legend based on your feedback, while removing the potentially confusing M1, M2, and M3 from the method introduction.

Reviewer 3 Report (New Reviewer)

Comments and Suggestions for Authors

The authors have provided appropriate revisions in response to the reviewer’s comments.
I have no further comments. 

Author Response

Commend1:The authors have provided appropriate revisions in response to the reviewer’s comments.I have no further comments.

Response1:Thank you so much for your time and effort in reviewing our manuscript again. We are delighted to hear that our revisions have addressed your concerns to your satisfaction. Your meticulous review and constructive feedback during both rounds of revision have been instrumental in strengthening the quality and clarity of our work.We are truly grateful for your expertise and dedication in helping us improve the manuscript.

This manuscript is a resubmission of an earlier submission. The following is a list of the peer review reports and author responses from that submission.

Round 1

Reviewer 1 Report

Comments and Suggestions for Authors

Thank you very much for the opportunity to review your manuscript “Nutrition Literacy among University Students in Beijing: Status, Determinants, and Implications”. Nutrition is crucial for health and well-being, especially during college, when young adults often face significant changes in their eating habits and lifestyle. Despite access to information, college students often lack adequate knowledge to make informed nutritional decisions, which can negatively affect their health and academic performance.
This article analyzes the nutrition knowledge of college students, focusing on their understanding of a balanced diet and the factors that influence this knowledge, such as education, social influences, and attitudes toward health. Identifying gaps in nutrition education is essential to promote healthier eating habits, improve students' overall health, and support their academic success.
I would like to thank the authors for their valuable contribution to this important area of research. However, some points deserve further discussion. Therefore, the following are my suggestions and observations to improve the clarity of the article.

2. Materials and Methods

2.1. Study design and Participants:

- How many people are part of the university community or what was the participation rate?

- How was information disseminated to recruit participants?

- What was the minimum/mean age for participation?

- How were the 53 pilot study participants selected? Did they meet the study's inclusion criteria?

2.2. Measures

With respect to the questionnaire, the category excellent knowledge is mentioned in several sections, which is then defined in the statistics as NL >80. Is this an arbitrary category or is it described in this way in the questionnaire?

2.3. Statistical Analysis

- line 129: categorical data with percentages and frequencies, quantitative variables as mean and standard deviation.

- t- test are also performed to compare scores among dimensions? (figure 2). It should be indicated.

- line 131: Authors mention “For correlation analysis, both generalized linear and logistic linear regression analyses”. Correlation and regression are not the same, please clarify these concepts. It also should be checked if generalized or general linear models are performed.

- line 134: To which analysis does this sentence correspond? “A two-tailed P < 0.05 will be considered statistically significant”

- line 146: Figure 1 should be included in the results.

3. Results

3.1. Demographic characteristics of the study participants

- Table 1: improve the presence of the table

- line 157: There is no “mean ± standard deviation” in the table

- line 160:  What does this sentence refer to? “Ways for families to acquire knowledge” and “Personal ways of acquiring knowledge” were defined as less if there were fewer than three ways, else defined as more.

- line 161: This paragraph should be in the discussion and rewrite: We can learn from this information that Chinese college students’ food skills, especially food selection skills, are not good enough, and that attaching importance to improving food selection skills is probably an effective way to improve people’s NL.

- line 168: It is not a correlation analysis.

- line 182: in supplementary tables frequencies and percentages should be included.

- line 237: improve the presence of the table

4. Discussion:

- line 303-307: Rewrite this paragraph

Supplementary information:

-          supplementary table S1: frequencies and percentages should be included.

-          supplementary table S2: gender, major… should be in capital letters

-          Part II: Nutrition Literacy Evaluation Scale: correct answers should be indicated.

Reviewer 2 Report

Comments and Suggestions for Authors

This cross-sectional study highlights important findings about the level of Nutrition Literacy among college students. Strengths include a validated survey tool, sample size based on a power calculation, and inclusion of students from multiple sites (12 universities). Additional information about the survey tool categorization and specifics to existing data used for mediating confounders and the NL variable is needed.

Specific comments follow.

 Abstract
“…this research utilized a comprehensive questionnaire to measure” – delete the word comprehensive

“…with only 7.6% of participants achieving an excellent rate.” – delete the word only

Introduction
“Additionally, increasing evidence suggests a significant correlation” - add statistic of significance or change word to considerable

“Nutrition literacy (NL) was first defined…” – this is the definition for health literacy, change to definition for Nutrition literacy, see this paper for a definition of NLL Foods. 2023 Jul 19;12(14):2751. doi: 10.3390/foods12142751

Materials and Methods

2.2 Measures                                                                 
The questionnaire design for this study was structured into two main sections to comprehensively – delete comprehensively

Regarding the General Population Nutrient Literacy Questionnaire, “well-established instrument” – is it a validated, if so, instead of well-established say validated

How is the General Population Nutrient Literacy Questionnaire scored? Are there specific category ratings that map to the participant scores? If so, please add.

2.3 Statistical Analysis
“Baron and Kenny's procedure will be used to evaluate...” change will be to was

Results

3.1. Demographic characteristics of the study participants
Suggest changing “We collected 823 answers of our questionnaire through the Internet, 765 of them con-formed to the quality inspection requirements, so the questionnaire efficiency was 93.0%, which meet with our expectation. In our study population, female was more than male, accounting for 63.66%, medical students 14.64%, freshmen 33.20%, and key universi-ties 63.79%, details were shown in Table 1.  

To: A total of 823 participants completed the online questionnaire, 765 conformed with quality inspection requirements; the questionnaire efficiency was 93.0%. In our study population, female was more than male, accounting for 63.66%, medical students 14.64%, freshmen 33.20%, and key universi-ties 63.79%, details were shown in Table 1.

Suggest changing: “According to the results of the study, the mean NL score was 66.74±9.07.”

To: The mean NL score for participating college students was 66.74±9.07. 

“The excellent rate under the comprehensive nutrition literacy was 7.6%.” If the General Population Nutrient Literacy Questionnaire does not define this as an “excellent” rate, remove the word “excellent.”  Do the same throughout the paper. Only add the adjectives before the rate scores if they are used as the scoring category for the questionnaire.

“In contrast, Student from importent school…”  Define what an important school is. I am not familiar with that phrase.

In terms of the NL score, compare to students with a monthly discretionary income below 2000 CNY, those with incomes ranging from 2000 Nutrients 2024, 16, x FOR PEER REVIEW 6 of 11

Suggest changing in all places it is used: “compare to students”
To: compared with students

Also, in this section, there is a very long, run on sentence which makes the information difficult to comprehend. Instead of semi-colons, break into sentences with periods. 

3.4. Mediating effect of NL on health behaviors
This study leverages existing data—what is the source of the existing data?

Comments on the Quality of English Language

One general concern is the use of adjectives which I have suggested removing to ensure the objectivity of the reporting on the study methods and results. The paper requires proofreading; the English language is some places needs tweaking. For example, ensuring the entire paper is in past tenses, correcting awkward phrasing, and eliminating run on sentences. I offer some edits in specific comments (see above).
